# Lafora Disease and Alpha-Synucleinopathy in Two Adult Free-Ranging Moose (*Alces alces*) Presenting with Signs of Blindness and Circling

**DOI:** 10.3390/ani12131633

**Published:** 2022-06-25

**Authors:** Madhu Ravi, Atilano Lacson, Margo Pybus, Mark C. Ball

**Affiliations:** 1Animal Health and Assurance, Alberta Agriculture, Forestry and Rural Economic Development, Edmonton, AB T6H 4P2, Canada; 2Department Laboratory Medicine and Pathology at Alberta Precision Laboratories, University of Alberta, Edmonton, AB T6G 2R3, Canada; atilano.lacson@albertaprecisionlabs.ca; 3Fish and Wildlife, Alberta Environment and Parks, Edmonton, AB T6H 4P2, Canada; margo.pybus@gov.ab.ca (M.P.); mark.ball@gov.ab.ca (M.C.B.); 4Department of Biological Sciences, University of Alberta, Edmonton, AB T6G 2E9, Canada

**Keywords:** moose, circling, blindness, Lafora disease, polyglucosan bodies, Lewy body, synucleinopathies, α-Synuclein

## Abstract

**Simple Summary:**

Reports of behavioral signs, such as blindness and circling in free-ranging moose from different parts of the world, have spurred comprehensive pathological investigation to find the causes of the disease that have clinical relevance. In this case study, brains collected from two adult free-ranging moose (*Alces alces*) cows that were seemingly blind and found walking in circles were examined by light and electron microscopy with further ancillary testing. Here, we report for the first time Lafora disease and alpha-synucleinopathy in two wild free-ranging moose cows who presented with abnormal behavior and blindness, with similar neuronal polyglucosan body (PGB) accumulations identified in humans and other animals. Microscopic analysis of the hippocampus of brain revealed inclusion bodies resembling PGBs (Lafora disease) in the neurons with ultrastructural findings of aggregates of branching filaments, consistent with polyglucosan bodies. Furthermore, α-synuclein immunopositivity was noted in the hippocampus, with accumulations of small granules ultrastructurally distinct from PGBs and morphologically compatible with alpha-synucleinopathy (Lewy body). The apparent blindness found in these moose could be related to an injury associated with secondary bacterial invasion; however, an accumulation of neurotoxicants (PGBs and α-synucleins) in retinal ganglion cells could also be the cause. Lafora disease and alpha-synucleinopathy were considered in the differential diagnosis of the young adult moose who presented with signs of blindness and behavioral signs such as circling.

**Abstract:**

Lafora disease is an autosomal recessive glycogen-storage disorder resulting from an accumulation of toxic polyglucosan bodies (PGBs) in the central nervous system, which causes behavioral and neurologic symptoms in humans and other animals. In this case study, brains collected from two young adult free-ranging moose (*Alces alces*) cows that were seemingly blind and found walking in circles were examined by light and electron microscopy. Microscopic analysis of the hippocampus of the brain revealed inclusion bodies resembling PGBs in the neuronal perikaryon, neuronal processes, and neuropil. These round inclusions measuring up to 30 microns in diameter were predominantly confined to the hippocampus region of the brain in both animals. The inclusions tested α-synuclein-negative by immunohistochemistry, α-synuclein-positive with PAS, GMS, and Bielschowsky’s staining; and diastase-resistant with central basophilic cores and faintly radiating peripheral lines. Ultrastructural examination of the affected areas of the hippocampus showed non-membrane-bound aggregates of asymmetrically branching filaments that bifurcated regularly, consistent with PGBs in both animals. Additionally, α-synuclein immunopositivity was noted in the different regions of the hippocampus with accumulations of small granules ultrastructurally distinct from PGBs and morphologically compatible with alpha-synucleinopathy (Lewy body). The apparent blindness found in these moose could be related to an injury associated with secondary bacterial invasion; however, an accumulation of neurotoxicants (PGBs and α-synuclein) in retinal ganglions cells could also be the cause. This is the first report demonstrating Lafora disease with concurrent alpha-synucleinopathy (Lewy body neuropathy) in a non-domesticated animal.

## 1. Introduction

Lafora disease (LD) is an autosomal recessive progressive neurodegenerative glycogen-storage disorder first documented in 1911 in human patients who presented with progressive myoclonus epilepsy (PME) [1,2]. LD in humans is associated with a mutation of *EPM2A* or *EPM2B/*
*NHLRC1* genes, which encode laforin and malin, respectively, resulting in aberrations of glycogen metabolism in nervous tissues. Laforin has a carbohydrate-binding module and a dual-specificity phosphatase domain and dephosphorylates glycogens. Malin is a ubiquitin ligase that ubiquitinates laforin and promotes its degradation. Malin forms a complex with laforin, and ubiquitinates enzymes that are involved in laforin-dependent glycogen synthesis. Laforin and malin are also involved in protein folding and protein degradation, and autophagy with the assistance of chaperone proteins, which are involved in the ubiquitin–proteasome system. Dysfunction of laforin or malin results in the accumulation of glycogens and glycosaminoglycans, resulting in the formation of polyglucosan or Lafora bodies [2,3,4,5,6].

The symptoms of LD usually appear during late childhood or adolescence (range: 8–19 years; peak: 14–16 years) in humans, and are characterized by early manifestations of focal visual seizures with transient blindness or neuropsychiatric symptoms, such as simple or complex visual hallucinations and behavioral changes. Generalized tonic-clonic seizures, drop attacks, and myoclonus typically occur at rest and increase with emotion, action, or photic stimulation. The symptoms then progress toward intractable stimulus-sensitive myoclonus, refractory seizures, psychosis, ataxia, blindness, and dysarthria [7,8]. The myocloni remain asymmetric and segmental but become almost constant, and massive myoclonic jerks, speech difficulties, ataxia, progressive dementia with apraxia, and visual loss appear with disease progression. Finally, affected individuals become totally disabled and death usually occurs within 10 years from the onset, often during an episode of status epilepticus with aspiration pneumonia [7,9,10].

A distinctive histopathologic change in LD is characterized by dense accumulations of malformed and insoluble glycogen molecules, known as polyglucosan bodies (PGBs). They differ from normal glycogens by lacking the symmetric branching that allows glycogens to be soluble. These periodic acid-Schiff (PAS)-positive and PAS diastase (PASD)-resistant PGBs are present in neurons of all brain regions, specifically in perikaryon, neuronal processes, and dendrites [7,10,11]. In addition to the brain, they are also seen in the periportal hepatocytes, skeletal and cardiac myocytes, and in the eccrine duct and apocrine myoepithelial cells of the sweat glands of humans [12,13,14]. Electron microscopy has confirmed the presence of asymmetric branching and fibrillary polyglucosan accumulations ranging from 50 to 125 nm that bifurcate at regular intervals in PGB inclusions [15,16,17].

Adult polyglucosan body disease (APBD) is another kind of heritable glycogen storage disease (GSD) often associated with glycogen-branching enzyme (GBE) deficiency, which causes a neurodegenerative disorder due to an accumulation of abnormal polyglucosans in the cells of various body systems of humans and horses [18,19]. Polyglucosan accumulations in APBD have a striking similarity to PGBs, which are also insoluble, lack the typical glycogen structure, and are resistant to α-amylase and diastase digestion [18,20].

Among non-human species, LD has been reported in cats, cows, dogs, gray-headed flying foxes, raccoons, fennec foxes, and parakeets [21,22,23,24,25]. However, genetic abnormalities with inheritance have been demonstrated only in certain breeds of dogs [26,27,28,29,30,31,32,33,34,35]. The key factor involved in neurodegeneration in Parkinson’s disease (PD) in humans is the accumulation of α-synuclein caused by mutations in the *SNCA* gene, which encodes α-synucleins. The abnormal accumulation and aggregation of α-synucleins in the form of Lewy bodies (LBs) and Lewy neurites (LNs) result in neurotoxicity, with the dysfunction and degeneration of neurons in PD patients. α-synucleins are misfolded and polymerized to form toxic fibrils upon binding to synthetic membranes or certain lipid surfaces, coalescing into pathologic inclusions in various neurodegenerative diseases such as Parkinson’s disease, Lewy body dementia, and multiple-system atrophy [36,37,38,39].

The occurrence of alpha-synucleinopathies is unknown in domestic and wild animals. Here, we report for the first time LD and alpha-synucleinopathy in two wild free-ranging moose who presented with abnormal behavior and blindness with similar neuronal PGB accumulations identified in humans and other animals.

## 2. Materials and Methods

Moose 1, a young adult (age: 2.5+ years) free-ranging moose cow, was found walking in circles in the middle of a quarter section of a farming land in northern Alberta, Canada. Moose 2, another adult (age: 2.5+ years) free-ranging moose cow, was found displaying odd behavior on a residential property 200 km southwest of the location described for Moose 1.

The field staff reported that both moose had visibly clouded eyes and were unresponsive to visual stimulation. Both moose were humanely euthanized by Fish and Wildlife enforcement staff due to clinical deterioration and public safety. The moose heads were collected, frozen, and submitted to Alberta Agriculture and Forestry for pathological examination and testing for various bacterial, viral, parasitic, and prion agents.

Eye swabs were collected for routine bacterial culture and polymerase chain reaction (PCR) testing for Herpesviridae genome, infectious bovine rhinotracheitis virus (IBRv), malignant catarrhal fever virus (MCFv), and Chlamydophila and Mycoplasma species. Obex and retropharyngeal lymph nodes were submitted to Chronic Wasting Disease (CWD) Surveillance Testing (ELISA) Program. The brain, trigeminal ganglion, and both eyes and eye lids were fixed in neutral 10% buffered formalin and routinely processed for histology using standard protocols. Tissue sections obtained from the eyes, eyelids, trigeminal ganglion, and brain, including the cerebellum, medulla oblongata, pons, midbrain at the level of the rostral colliculi, thalamus, hippocampus, basal nuclei, and cerebrum were stained with hematoxylin and eosin (H&E) using the standard protocol. Special stain tests, such as PAS, PASD, Grocott’s methenamine silver (GMS), and Bielschowsky’s staining were performed as previously described. [40] Immunohistochemistry for α-synucleins was performed by Alberta Precision Laboratories using their standard protocol with a mouse monoclonal antibody to α-synuclein (PGB 509 1/75; Abcam Inc., Toronto, ON, Canada). Electron microscopy on the hippocampus of the brain was performed as previously described by Sergi et al. [41]

## 3. Results

Both moose tested negative for CWD with ELISA. No significant bacterial pathogens were isolated from the bacterial cultures obtained from eye swabs. Both moose tested negative for Herpesviridae genome, IBRv, MCFv, and Mycoplasma species. The eye swab specimen of Moose 1 tested positive for Chlamydia pecorum, but that of Moose 2 tested negative for any Chlamydia species.

At the post-mortem inspection, both moose exhibited gross evidence of bilateral corneal opacity and ulceration. No gross pathologic lesions were observed in the brains and middle ears of either moose. Gross visual search of appropriate tissues did not reveal any evidence of parasites, including meningeal worm (*Parelaphostrongylus tenuis*), eyeworm (*Thelazia* sp.), or carotid worm (*Elaeophora schneideri*).

There was no histological evidence of inflammation to suggest that there was an infectious disease present in the brain tissues. Microscopic lesions were predominantly confined to the hippocampus; neuronal cell bodies, axons, dendrites, glial cells, and astrocytes multifocally contained large (up to 50 microns in diameter), round, basophilic to amphophilic globular inclusions (Figure 1). Extensive lesions were noted in CA1, CA2, CA3, dentate neurons, the subdentate layer, and the subiculum of the hippocampus of Moose 1. Lesion were less severe in Moose 2, involving the CA1 and CA2 fields of the hippocampus (Table 1). The nerve cell bodies were enlarged with most nuclei compressed or displaced to the periphery due to an abnormal accumulation of PGBs. The inclusions had a homogeneous core with a faintly staining radiating periphery. They were positively stained with PAS (Figure 2) and GMS (Figure 3) and were PAS diastase-resistant (PASD) (Figure 4 a and b) which also accentuated their structures. PGBs also showed moderate-to-faint positive staining with a Bielschowsky stain (Figure 5; Table 1).

Ultrastructurally, the PGB inclusions in CA1, CA2, and CA3 of the hippocampus were non-membrane-bound electron-dense bodies (Figure 6) composed of accumulations of osmophilic orderly branching filamentous materials (Figure 7) with scattered dense aggregates of glycogen-like material at the peripheries of some inclusions (data not shown).

The results of immunohistochemistry using α-synucleins demonstrated numerous intra- and extra-cellular protein aggregates that positively stained with the neurons, glial cells, neuronal process of CA3 and CA4, subiculum, and white matter of the hippocampus of Moose 1 (Table 1; Figure 8). Minimal staining was noted in the subiculum and white matter of the hippocampus Moose 2. (Table 1). PGBs were consistently negative for α-synuclein antibody staining (Table 1). Ultrastructural examination of coarsely granular material stained positive with a Bielschowsky stain, and α-synuclein antibodies (as shown in Figure 8) in the subiculum; white matter of the hippocampus showed membrane-bound inclusions filled with degenerate organelles interspersed with electron dense bodies (neuromelanosomes), lipid bodies, vesicles, and distorted mitochondria (Figure 9). α-synuclein-positive areas were negatively stained in a GMS test (data not shown).

## 4. Discussion

Lafora disease (LD) is a rare autosomal recessive disease first documented in humans in 1911, and is commonly reported in humans from various parts of the world where a high rate of consanguinity exists [7,10]. LD is characterized by dense accumulations of abnormal and insoluble glycogen molecules, known as polyglucosan bodies (PGBs) [7,10]. LD patients typically exhibit increased neuronal cell death and neurologic symptoms due to progressive accumulation of PGBs in the brain, resulting in neurotoxic degenerative disease [42]. Behavioral and electrophysiological evidence supports the hypothesis that the hippocampus plays a role in processing spatial information and orientation. As the hippocampus is part of the limbic system, any lesion affecting this structure likely disrupts spatial and visual orientation [43,44,45,46]. In this study, the hippocampi of two moose were severely affected with PGB accumulations, and the staining properties of these polyglucosan accumulations in both moose were similar to reports in other species [25,47,48]. The ultrastructural morphology of these non-α-synuclein immunoreactive inclusions in the moose of this study was similar to those previously described in other species [16,21,24,49,50].

The exact mechanism of regulation of glycogen metabolism by the laforin–malin system in the generation of PGBs remains unknown. The transformation of spherical and soluble glycogens to insoluble misstructured polyglucosans in the neurons and astrocytes due to mutations in the *EPM2A* or *EPM2B* genes is thought to be the pathogenesis underlying LD in humans and animals [26,28,34,51,52]. However, it was suggested that glycogen synthase is the only enzyme that catalyzes formation of α1 ± 4 interglucosidic linkages that generate glycogen (or polyglucosan) strands [53]. Different studies performed on LD-infected mice have indicated that a reduction or blockage in glycogen synthase activity eliminates PGBs and the neurological phenotype of the disease [54,55]. A higher prevalence of LD with an *Epm2B* mutation was reported in mature miniature wirehaired Dachshund dogs who commonly presented with signs of panic attacks, reflex and spontaneous myoclonus hypnic myoclonus, dementia, impaired vision and blindness, aggression, deafness, and fecal/urinary incontinence. Over one-third of the dogs with an *EPM2B* genetic defect presented with poor vision/blindness 1 to 3 years following initial clinical evaluation. The researchers hypothesized that the dogs may have experienced frightening visual hallucinations, as the dog’s behavior suggested fear and the seeking of an escape response [56]. Out of 250 human patients described with LD to date, 42% were caused by *EPM2A* mutations and 58% by *EPM2B* mutations [52].

Histologically, in addition to the brain, cytoplasmic PGBs have been discovered in the spinal cord, sweat glands of skin, liver, and cardiac and skeletal muscles in animals and humans [7,11,12,35]. Visual impairments with retinal alterations were reported in human patients with LD [57,58]. No inclusions were noted in the ocular muscles in our study, and the other body systems of these moose were not evaluated as they were unavailable. Other studies have indicated that an accumulation of PGBs in non-nervous tissues typically does not affect their physiologic function [7,56].

Listeriosis and thiamine deficiency as causes of blindness and circling were ruled out due to a lack of associated histopathological lesions in the brain tissues. Blindness was a clinical presentation of these moose, and they had gross evidence of ulcerative keratitis with histological evidence of chronic inflammation in the corneas. We suspect trauma as a cause of corneal lesions with secondary bacterial infections, as these moose were unable to orient themselves due to the lesions in their hippocampi. These disoriented moose may have run into trees or bushes that then traumatized their eyes and resulted in blindness. *Chlamydia pecorum* was identified from the eye swab collected from one of the moose, and we suspect it as a secondary invader. Although reports indicated that *Chlamydia pecorum* was detected from animals with keratoconjunctivitis [59], its role in the pathogenesis of keratoconjunctivitis in animals is undetermined [60]. Additionally, retinal accumulations of these neurotoxicants in the retinal ganglion cells cannot completely be ruled out as a cause of blindness in these animals because tissue autolysis and freezing and thawing artifacts prevented a complete histological evaluation of the eyes.

Lafora and polyglucosan diseases require differentiation from inflammatory diseases of the brain and other storage disorders, such as glycogen storage disease type IV (GSD IV), glycogen branching enzyme deficiency disease (GBSD), and adult polyglucosan body disease (APBD) caused by mutations of the *GBE 1* gene, based on the morphological features of inclusions in brain tissues [61]. The prominent lesion in GSD IV is in the skeletal muscles of affected individuals, which we did not see in this case study. Moreover, the inclusions were ultrastructurally different than those of the GSD IV disease reported in humans and horses. Another differential to consider is the non-pathogenic morphologically similar corpora amylacea. These structures can occur as an aging-related change in many species, including cats, dogs, and humans. Corpora amylacea are not usually present in neurons but occur in glia or freely in the neuropil, and these accumulations are not usually associated with the neurologic diseases associated with blindness or behavioral signs. There was no evidence of inflammation or abscessation in the brains of these moose.

In addition to PGB accumulations, α-synuclein immunoreactive accumulations were also detected in the hippocampus region of the moose in this study. These accumulations are morphologically distinct from the PGBs found in the hippocampi of these moose. α-synucleins are a pathological hallmark of Lewy body associated disorders, such as Parkinson’s disease (PD) and dementia with Lewy bodies in humans [36,37]. α-synucleins are endogenous proteins encoded by the *SNCA* gene. Mutations in this gene cause an accumulation of insoluble α-synuclein protein inclusions known as Lewy bodies, which accumulate in the neurons and neurites, glia, and presynaptic terminals of the brain [36,37]. Diseases associated with an accumulation of α-synuclein-positive materials in the brain are known as alpha-synucleinopathies [62,63,64]. Progressive deposition of α-synuclein inclusions in distinct brain areas such as the hippocampus, cortex, and amygdala result in LB-associated neurologic disorders in humans [65]. Inhibition of the autophagy–lysosome pathway is the proposed mechanism by which α-synucleins contribute to the pathogenesis of LB-associated neurodegenerative diseases with an accumulation of α-synuclein fibrils resulting in the failure to remove insoluble aggregates due to dysfunctional autophagy, increased oxidative stress, and mitochondrial dysfunction [4,36,38]. Ultrastructurally, LBs primarily consist of a crowded membranous medley of vesicular structures and dysmorphic organelles [62,63,64,66], similar to those accumulations seen in this study.

Neuromelanin (NM)-containing organelles, as found in α-synuclein immunoreactive areas in this case study, have a very slow turnover during the life of neurons in PD due to reduced enzymatic activity and likely impaired capacity for lysosomal and autophagosome fusion [67]. The interaction of NM with α-synucleins may be a mechanism for this pigment to modulate neuronal vulnerability, and α-synuclein is over-expressed in individual melanized neurons, likely resulting in a vicious cycle of mutual promotion and eventual neuronal pathology in PD [68,69] Overall, these deposits are potentially neurotoxic, as these protein accumulations can interfere with the cell’s ability to prevent protein misfolding, revert misfolded proteins to normal, or eliminate misfolded proteins by degradation or autophagy, constituting the basis for pathological changes.

## 5. Conclusions

This is the first report of Lafora disease in a non-domestic species with concurrent accumulations of PGBs and α-synucleins. Diseases associated with alpha-synucleinopathies have not been reported in domestic or wild animal species. The significance of finding α-synuclein reactivity in the brains of these animals is unknown. Currently, in the literature, there are no published reports describing an accumulation of PGBs in moose, cervids, or any domestic or wild animals. Based on the information available from other species, we suspect mutations of genes encoding laforin/malin and α-synucleins due to inbreeding as a possible mechanism for the manifestation of clinical signs and pathological lesions. Future studies are needed to confirm the underlying cause of these neurodegenerative diseases through the genetic testing of genes encoding laforin/malin and α-synucleins. Moreover, prospective epidemiological analyses are needed to further characterize these neurodegenerative diseases in moose.

## Figures and Tables

**Figure 1 animals-12-01633-f001:**
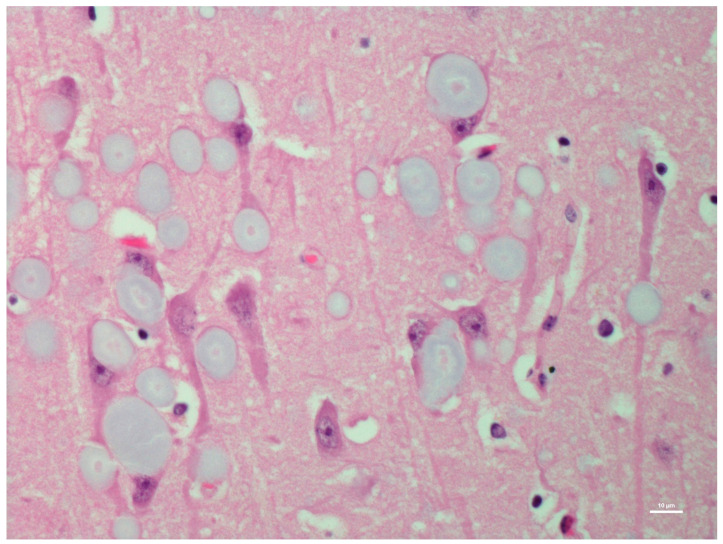
Hematoxylin and eosin (H&E)-stained section of hippocampus region of the brain of a moose showing round polyglucosan bodies (PGBs) in the perikaryon, neuronal processes, and neuropil. Note the outer pale layer and basophilic core.

**Figure 2 animals-12-01633-f002:**
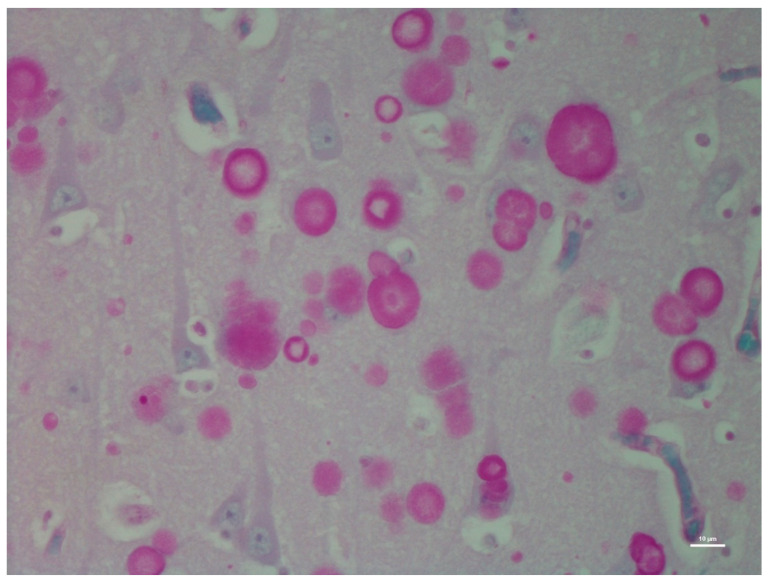
Periodic acid-Schiff (PAS)-stained section of the hippocampus region of the brain of the same moose showing round polyglucosan (Lafora) bodies.

**Figure 3 animals-12-01633-f003:**
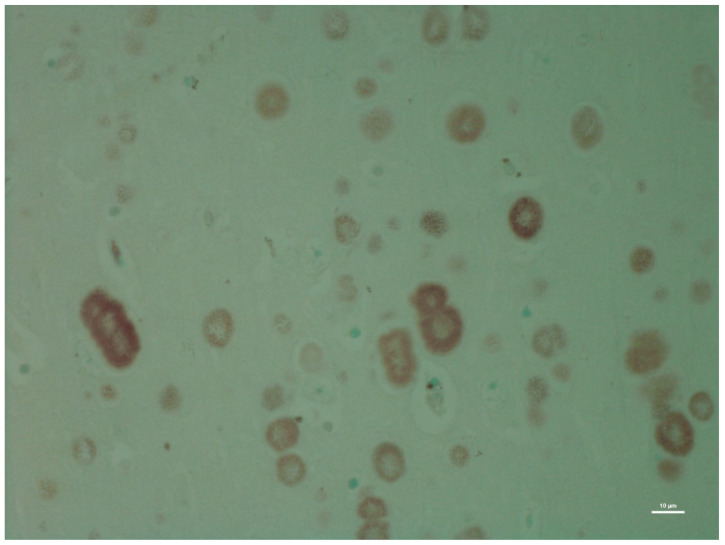
Grocott’s methenamine silver (GMS)-stained section of the hippocampus of the same moose showing dark-stained polyglucosan (Lafora) bodies.

**Figure 4 animals-12-01633-f004:**
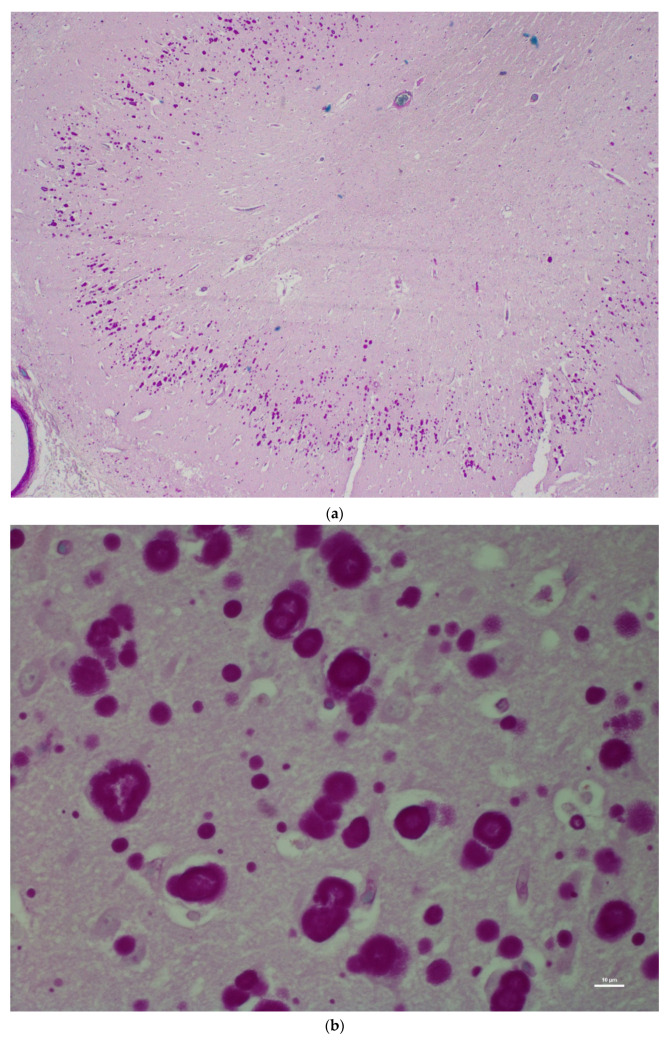
(**a**) Low magnification (2x) showing a diastase-digested periodic acid-Schiff (PASD) section of the hippocampus of the same moose showing polyglucosan (Lafora) bodies; (**b**) higher magnification showing a diastase-digested periodic acid-Schiff (PASD) section of the hippocampus of the same moose showing polyglucosan (Lafora) bodies.

**Figure 5 animals-12-01633-f005:**
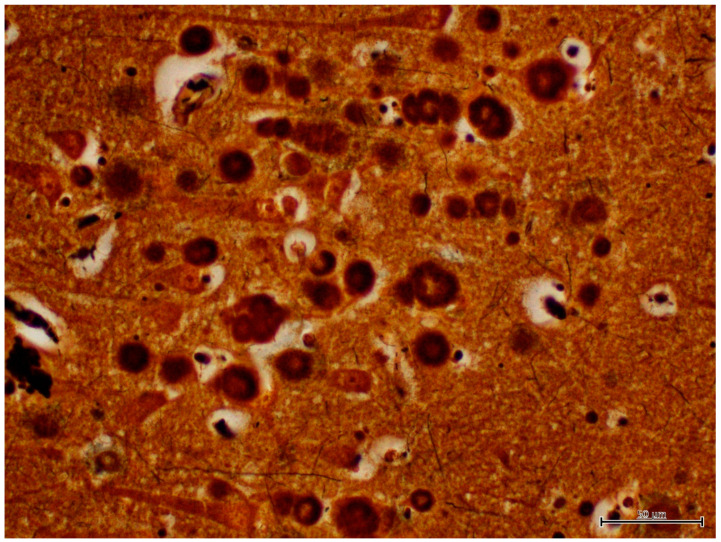
Bielschowsky’s silver-stained section of the hippocampus of the same moose showing dark-stained polyglucosan (Lafora) bodies.

**Figure 6 animals-12-01633-f006:**
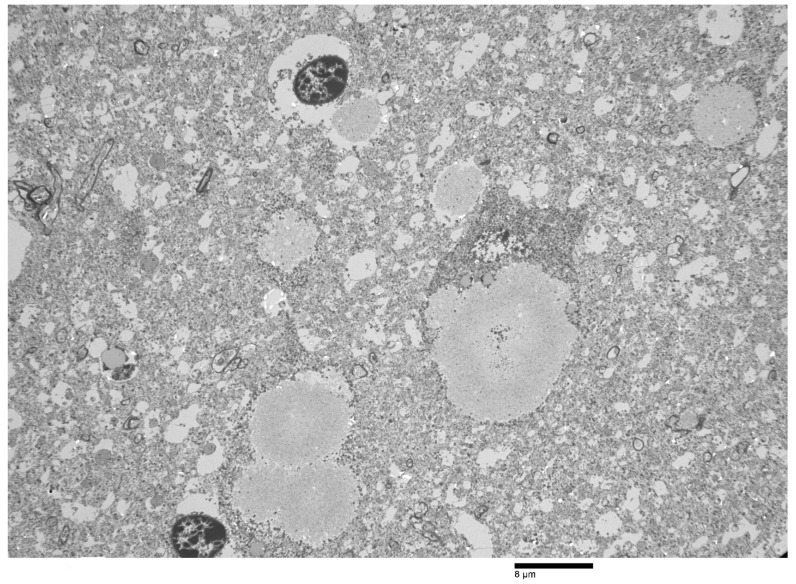
Low-power electron micrograph of the perikaryon with polyglucosan bodies (PGBs) in neurons of the hippocampus of the same moose. Electron-dense core is the equivalent of the basophilic core seen in H&E-stained material.

**Figure 7 animals-12-01633-f007:**
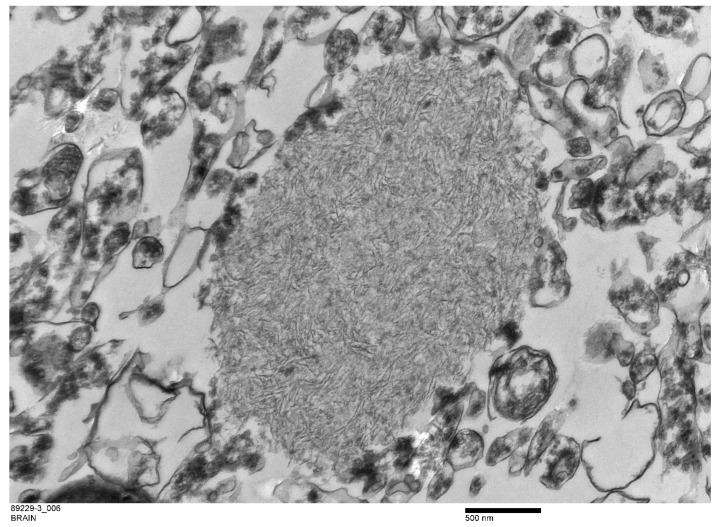
Ultrastructure image showing branched filamentous material in a non-membrane-bound polyglucosan body (PGB).

**Figure 8 animals-12-01633-f008:**
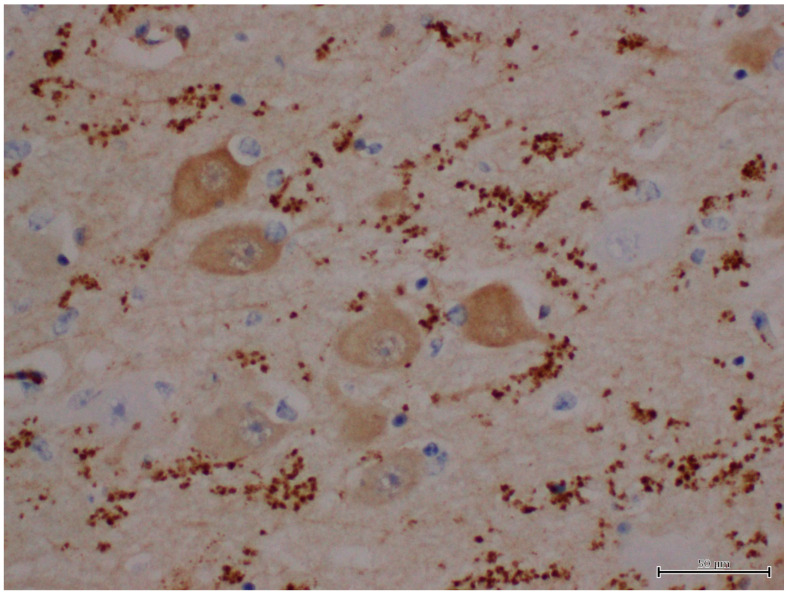
IHC labeling using α-synuclein monoclonal antibodies to label α-synucleins in the neuropil of the hippocampus.

**Figure 9 animals-12-01633-f009:**
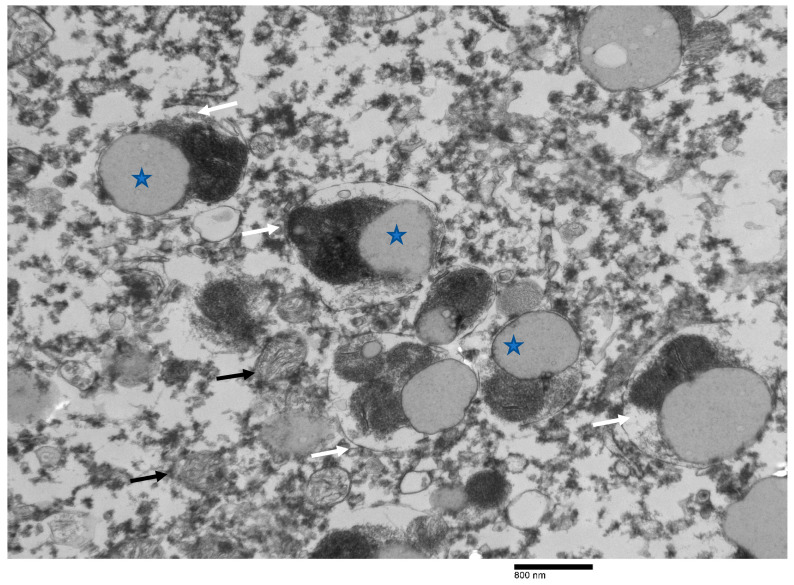
Ultrastructure image from the positively labeled area of the hippocampus with α-synuclein monoclonal antibodies showing electro- dense membranous inclusions (white arrows). Vesicles, lipid bodies (stars), and distorted mitochondria are also interspersed with the inclusions (black arrows).

**Table 1 animals-12-01633-t001:** Lesions with severity in various regions of the hippocampus identified in light and electron microscopy.

		H&E		PAS		PASD		Bielschowsky		α-Synuclein		Ultrastructure	
Study Subject	Hippocampus Region	PGB	LB	PGB	LB	PGB	LB	PGB	LB	PGB	LB	PGB	LB
Moose 1	Hippocampus												
	CA1	+++	−	+++	−	+++	−	+++	−	−	±	+++	+
	CA2	+++	−	+++	−	+++	−	+++	-	−	+	+++	+
	CA3	+++	−	+++	−	+++	−	+++	+	−	+++	+++	+
	CA4	−	−	−	−	−	−	−	+	−	+++	−	−
	Dentate neurons	+++	−	+++	−	+++	−	+++	−	−	−	+++	−
	Subdentate layer	+++	−	+++	−	+++	−	+++	−	−	−	+++	ND
	Subiculum	+++	−	+++	−	+++	−	+++	−	−	++	+++	ND
	White matter	−	−	−	−	−	−	−	−	−	++	−	ND
Moose 2	Hippocampus												
	CA1	+	−	+++	−	+++	−	++	-	−	−	++	−
	CA2	+	−	+++	−	+++	−	++	-	−	−	++	−
	CA3	−	−	−	−	−	−	−	−	−	−	−	−
	CA4	−	−	−	−	−	−	−	−	−	−	−	−
	Dentate neurons	−	−	−	−	−	−	−	−	−	−	−	++
	Subdentate layer	−	−	−	−	−	−	−	−	−	−	−	++
	Subiculum	−	−	−	−	−	−	−	−	−	+	−	++
	White matter	−	−	−	−	−	−	−	−	−	+	−	++

H&E—hematoxylin and eosin; PAS—periodic acid-Schiff; PASD—periodic acid-Schiff diastase; GMS—Grocott’s methenamine silver; PGB—polyglucosan body; LB—Lewy body; + (minimal); ++ (mild); +++ (moderate); – (negative); and ND (not done)

## Data Availability

Not applicable.

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
