# Peer review of "Lafora Disease and Alpha-Synucleinopathy in Two Adult Free-Ranging Moose (Alces alces) Presenting with Signs of Blindness and Circling"

_animals, 2022, doi:10.3390/ani12131633_

Round 1
Reviewer 1 Report
Please see attached.

Author Response
General comments/recommendations:
Point 1. I assume thiamine deficiency was ruled out based on lack of lesions. Likewise, listeria was not
specifically mentioned but was presumably ruled out by the lack of encephalitis.
Response 1: Added “Listeriosis and thiamine deficiency as a cause of blindness and circling were ruled out due to lack of associated histopathological lesions in the brain tissues.” (lines: 261-262)
Point 2: Lines 119-123 in the Material and Methods actually discuss gross necropsy findings (Results).
Response 2: Moved lines 119-123 to Results
Point 3: Lines 112-113 are similar in that regard but do explain the reason for euthanasia.
Response 3: Deleted “They were however responsive to auditory stimulation.” Added ” due to clinical deterioration and public safety.” to the line 113
Point 4: As the authors mention in the Discussion (lines 261-269), it is highly unlikely that the apparent blindness described was due directly to Lafora Disease and/or Alpha-Synucleinopathy but is probably secondary to injury or other infection. This should probably be made clear in the summary and abstract.
Response 4: Added “The apparent blindness described in these moose could be related to injury with bacterial invaders, however, accumulation of neurotoxicants (PGBs and α-synuclein) in retinal ganglions cells cannot be completely ruled out.” to summary (lines:27-29) and abstract (lines:50-52)
Point 5: There are several excellent photomicrographs!
Response 5: Thank you.
Specific comments/recommendations:
Point 6: Lines 16-17 seem to imply that there is only one (and previously unknown at that) cause for circling and blindness in moose. Of course, there are many. I think the authors might be trying to state that there is a need for investigations into undiagnosed or undescribed causes of these signs.
The fact that these were both cow moose could be mentioned in the abstract.
Are there any estimations of the age of these moose? I know precise estimates are difficult, but some idea of age-range would be helpful. It is stated that they are “young”, but no age range is given.
Response 6:
Added, “cows” to young moose in the summary and abstract wherever necessary.
Unfortunately, we did not collect the teeth for accurate age estimation since it is not a common practice. Based on dentition, we can say the age of cows as 2.5+ years. So added 2.5+ years to the methods. (lines: 117 & 119)
Point 7: Lines 126-127 and 142: What specific test was used to rule out CWD (IHC, ELISA)?
Response 7: ELISA was used. Added “ELISA” to lines 129 & 144.
Point 8: Line 145: The species is listed as “Chlamydophila pecorum” vs on line 266 & 268 it is refered to as “Chlamydia pecorum”. I believe the latter is now correct.
Response 7: Corrected as “Chlamydia pecorum”. Thank you.
Point 8: Lines 326-327: The authors mention that future studies should include genetic testing. Please be more specific. Are you suggesting genetic testing to look for inbreeding or for other susceptibility to these conditions
Response 8:
Added “…..due to inbreeding…” line 334
Implied genetic testing to find out mutations as a cause of genetic susceptibility to Lafora and Lewy body accumulations. Edited for clarity by mentioning them. (Lines 335 & 336)
Reviewer 2 Report
The behavioral and pathological picture is consistent with Lafora disease. Co-presence of the synucleopathy is highly intriguing. The paper would have been so much the stronger had the authors carried out the confirmatory genetic studies that they highlight by italicizing at the end of the manuscript. This reviewer would be happy to interact with them on this.
The Lafora and Lewy bodies are so clear that control images from aged moose brain are probably not needed, but this reviewer is not an expert in this and related species. Mention that these types of bodies to this extent are not seen in this species, with a reference, could help.
A lower magnification image encompassing the totality of the hippocampus would be useful. Also an image from the cortex showing presence (or total absence?) of Lafora bodies would be helpful.
Author Response
Point 1: Mention that these types of bodies to this extent are not seen in this species, with a reference, could help
Response 1: line 340-342 - added " As per the current literature search, there are no published reports indicating the accumulation of PGBs in moose or cervids, and α-Synuclein in any domestic or wild animals. "
Point 2: A lower magnification image encompassing the totality of the hippocampus would be useful. Also an image from the cortex showing presence (or total absence?) of Lafora bodies would be helpful.
Response 2: Added low magnification showing the extent of accumulations in the hippocampus in Figure 3a
Point 3: The paper would have been so much the stronger had the authors carried out the confirmatory genetic studies that they highlight by italicizing at the end of the manuscript. This reviewer would be happy to interact with them on this.
Response 3: Thank you for offering help. We are interested to collaborate. Please contact me my email: mbrdacvp@gmail.com
Reviewer 3 Report
The paper is very well written and sound in terms of strategy and methodology used. I do not have much to add to this interesting manuscript. What I suggest to the Authors is to consider the opportunity to have more animals to include in the study and an accompanying immunohistochemical support to the immunofluorescent analysis data that they obtained. In fact, α-synuclein antibodies work very well in immunofluorescent that is usually used in biopsies of patients with Parkinson disease at any phase of the clinical course. That apart, this study is neat and very important for the veterinary community.
Author Response
Point 1: The paper is very well written and sound in terms of strategy and methodology used. I do not have much to add to this interesting manuscript. What I suggest to the Authors is to consider the opportunity to have more animals to include in the study and an accompanying immunohistochemical support to the immunofluorescent analysis data that they obtained. In fact, α-synuclein antibodies work very well in immunofluorescent that is usually used in biopsies of patients with Parkinson disease at any phase of the clinical course. That apart, this study is neat and very important for the veterinary community.
Response 1: Thank you for your comments. Agree with the reviewer's comments. A study by having more animals and IHC analysis of data would be the future possibility when we find similar findings in more animals. Thank you again.